# The Evolving Landscape of Neoadjuvant Immunotherapy in Gastroesophageal Cancer

**DOI:** 10.3390/cancers16020286

**Published:** 2024-01-09

**Authors:** Colum Dennehy, Alisha F. Khan, Ali H. Zaidi, Vincent K. Lam

**Affiliations:** 1Sidney Kimmel Comprehensive Cancer Center, Johns Hopkins University School of Medicine, Baltimore, MD 21287, USA; cdenneh3@jh.edu; 2Allegheny Health Network Cancer Institute, Allegheny Health Network, Pittsburgh, PA 15212, USA; alishafkhan@icloud.com

**Keywords:** immunotherapy, esophagogastric cancer, gastric cancer, esophageal cancer, localized, perioperative, adjuvant, neo-adjuvant, resectable

## Abstract

**Simple Summary:**

Esophagogastric cancer remains a devastating diagnosis, even when it is caught early the likelihood of its recurrence remains significant. Treatment prior to surgery, or neoadjuvant therapy, has been a mainstay of management for esophagogastric cancers over the past decade. In this review we discuss how this treatment paradigm has evolved and the trials and pertinent results that shape our management today. Unlike traditional methods that focus solely on surgery, radiotherapy or chemotherapy, immunotherapy uses drugs that stimulate the body’s immune system to fight cancer. By administering these drugs before surgery, clinicians aim to shrink tumors, making them easier to remove, and potentially preventing cancer from spreading or returning in the future. We discuss recent studies investigating its use before surgery that have shown promising results indicating improved outcomes and reasonable safety. Although more research is needed to fully understand its long-term benefits, neoadjuvant immunotherapy represents a hopeful advancement in the quest for more effective treatments against this type of cancer.

**Abstract:**

Despite advances in treatment strategies and surgical approaches in recent years, improving survival outcomes in esophagogastric cancer (EGC) patients treated with curative intent remains a significant area of unmet need. The recent emergence of adjuvant immunotherapy as the standard of care for resected EGC demonstrates the impact of immunotherapy in improving recurrence-free survival. Neoadjuvant and perioperative immunotherapies represent another promising approach with potential advantages over adjuvant therapy. Despite the promising results of early neoadjuvant immunotherapy studies, there are several challenges and future research needs. The optimal timing, duration and number of doses in relation to surgery and the optimal combination of immunotherapies are still unclear. In addition, rigorous correlative studies need to be performed to identify biomarkers for patient selection and treatment response prediction to maximize the benefits of neoadjuvant immunotherapy. In this review, we provide a concise summary of the current standard of care for resectable EGC and discuss the rationale for the use of immune checkpoint inhibitors in this setting and the pre-clinical and early clinical data of these novel therapies. Finally, we will examine the potential role and future direction of immunotherapy in the treatment paradigm and the perceived challenges and opportunities that lay ahead.

## 1. Introduction

The upper gastrointestinal tract is sometimes regarded as a single entity; however, there is significant heterogeneity in tumor location, histology, and molecular characterization amongst global populations. Esophagogastric cancer (EGC) includes gastric (GC), esophageal (EC) and gastroesophageal junction (GEJ) cancers and represent a significant global public health concern as they are some of the fastest growing cancers in Western society and are among the leading causes of cancer death worldwide [1,2]. EGC remains a devastating diagnosis with a global incidence of approximately 1.6 million cases with associated poor survival rates regardless of histological subtype or tumor location [3]. Due to their inherent aggressive disease biology, the 5-year overall survival (OS) for resectable patients remains low at only 27–48% in esophageal cancer and 32–72% in gastric cancer [4,5]. This is likely due to the presence of micro-metastatic disease at the time of presentation and diagnosis. Population analyses of the United States have demonstrated a nearly 2.5-fold increased incidence of EGC over the past 50 years, which is likely correlated to the obesity epidemic, and this phenomenon will only worsen in the coming decades [6]. Over the past two decades, due to their relative rarity in Western populations, EGCs have been grouped in varying different combinations within the eligibility criteria of prospective randomized clinical trials. This has introduced an additional level of complexity, further challenging the extrapolation and practical implementation of their findings. In this review article, we will discuss the current clinical practice paradigms, treatment guidelines and the evidence supporting the multimodality approaches utilized in resectable, locally advanced EGC, discuss the rationale and data for neoadjuvant Immune checkpoint inhibitors (ICIs) in all the tumor subgroups, highlighting their respective merits and potential drawbacks and explore any novel biomarkers that could be incorporated into future treatment algorithms.

## 2. Current Practice and Standard of Care

Multimodality treatment with either neoadjuvant chemoradiotherapy (CRT) or perioperative triplet chemotherapy followed by surgery remains the standard curative approach in the treatment of resectable, locally advanced EGC and has been widely adopted in standard treatment for these patients [7,8,9]. Typically, clinicians consider CRT in esophageal cancer (particularly the squamous cell carcinoma (SCC) population) and GEJ cancer (Siewert I/II), whereas perioperative triplet chemotherapy is utilized for more distal tumors (Siewert III GEJ and gastric cancer), although these approaches may vary by institution and region. A comprehensive understanding of the various therapeutic strategies is necessary for prudent clinical utilization of the data, as each approach carries its own unique intricacies and constraints.

The purpose of any neoadjuvant treatment is to downstage patients prior to surgical resection and improve optimal resection rates (R0), therefore improving survival outcomes overall. Pathological regression in response to neoadjuvant therapy is an early indicator of therapeutic efficacy. Standard measurements are pathological complete response (pCR), defined as no viable tumor within the resected specimen or major pathological response (MPR), defined as 10 percent or less of residual viable tumor cells in the resected primary tumor or lymph nodes sampled. Other metrics utilized include the Becker and Mandard Tumor Regression Grades (TRG) 1 to 3 and 1 to 5, respectively. Mandard’s TRG 1–2 is equivalent to Becker’s 1a/1b, and they are thought to be equivocal as TRG areas under the ROC curve (AUCs) for 5-year survival are 0.71 and 0.72, respectively [10]. pCR after neoadjuvant treatment has been used as a surrogate endpoint for survival in other solid tumors such as early-stage breast cancer, Non-Small Cell Lung Cancer (NSCLC), melanoma and microsatellite instability-high (MSI-H) rectal cancer [11,12,13,14]. However, the reliability of pCR as a surrogate endpoint in esophageal cancer has not been demonstrated and is perhaps strongest in the SCC population, where the risk of distant metastasis is less than for adenocarcinoma [15]. The primary concern clinicians have with any neoadjuvant therapy is that patients may either become too unwell due to treatment toxicity or their cancer may progress during neoadjuvant therapy. In both scenarios, these patients may have theoretically missed a window of opportunity for curative surgery.

The established standard of care paradigm for most localized esophageal cancer is trimodality therapy with neoadjuvant CRT, delivering couplet chemotherapy, either carboplatin and paclitaxel (CROSS trial) or 5-fluorouracil based regimens with platinum, concurrently with radiation-followed-by-planned esophagectomy [7,16]. Both the CROSS and NEOCRTEC5010 trials demonstrated early efficacy with significantly higher R0 resection rates and pCR rates, with particular efficacy demonstrated amongst the SCC population (CROSS: 49% versus 23% (*p* = 0.008), NEOCRTEC5010: 43.2%) [7,17]. As expected, the magnitude of the benefit of chemoradiotherapy was greater in the radiosensitive SCC population, with OS rates at 10 years in the intervention arm at 46% (95% CI: 33–64) for patients with SCC and 36% (95% CI: 29–45) for patients with adenocarcinoma (AD) [18]. NEOCRTEC5010 only enrolled SCC patients, and they were treated with cisplatin/vinorelbine chemotherapy. It also demonstrated improved disease-free survival (DFS) and overall survival (OS) rates when compared to surgery alone [17]. Similar results were found with concurrent FOLFOX (5-FU, leucovorin, oxaliplatin), achieving a pCR rate of 28% and a 3-year OS rate of 45% in patients with resectable esophageal adenocarcinoma [16,19].

Initial data from the phase II PROTECT-1402 trial comparing carboplatin/paclitaxel vs. FOLFOX demonstrates similar resection (92.0%, CI: 80.8–97.8% vs. 87.5%, CI: 74.8–95.3%) and response rates (Tumor Regression Grade (TRG) 1–2 60.4%, CI: 44.3–74.2% vs. 44.2%, CI: 29.1–60.1%). Unexpectedly, post-operative complications were found to be significantly higher in the carboplatin/paclitaxel arm (43.8%, CI: 29.5–58.8% vs. 30.2%, CI: 17.2–46.1%) [20]. The Scandinavian NeoRes1 trial examined the role of neoadjuvant chemotherapy alone (5-FU, leucovorin, cisplatin) against the standard of neoadjuvant CRT for patients with esophageal or GEJ cancer (Siewert types I and II) and demonstrated that chemoradiation improved tumor regression and pCR (8% versus 24% (7/91 versus 22/90), *p* = 0.002), improved R0 resection rate (74% versus 87% 58/78 versus 68/78, *p* = 0.042), and lowered the frequency of lymph-node metastases compared to chemotherapy alone (39% vs. 64%), however, 5-year OS was equivocal at 42.2% (CI: 31.9–52.1%) in the chemoradiotherapy group versus 39.6% (CI: 29.5–49.4% *p* = 0.60) in the chemotherapy group [21]. One concern with CRT is the potential increase in post-operative morbidity and early mortality primarily as a result of radiation-induced complications such as post-operative pneumonia, chylothorax, anastomotic fistula, hemorrhage or leakage. In NeoRes1, the post-operative complication rate leading to death was statistically higher in the chemoradiotherapy group compared to the chemotherapy-alone arm (9% versus 1%, 8/90 versus 1/90, *p* = 0.02) [21]. Each of these chemotherapy regimens is included in the National Comprehensive Cancer Network (NCCN), American Society of Clinical Oncology (ASCO) and European Society for Medical Oncology (ESMO) guidelines for chemoradiation in esophageal cancer [22,23,24]. Additionally, the current practice for patients with resected esophageal or GEJ cancer with residual pathological disease (non-ypT0N0) after neoadjuvant CRT is adjuvant nivolumab (PD-1 inhibitor) for one year based on the CheckMate-577 trial [25].

In contrast, the dominant strategic approach adopted in localized gastric cancer (Stage Ib–III) is perioperative chemotherapy, both neoadjuvant and adjuvant chemotherapy. This model of treatment was first established after two major European trials, the UK Medical Research Council Adjuvant Gastric Infusional Chemotherapy (MAGIC) trial and the French Federation Nationale des Centres de Lutte contre le Cancer (FNCLCC)/the Federation Francophone de Cancerologie Digestive (FFCD) ACCORD trial [8,26]. The chemotherapy regimen was three cycles of neoadjuvant and adjuvant chemotherapy with epirubicin, cisplatin and 5-FU (ECF). The interventional arm demonstrated a significant improvement in 5-year OS (35% versus 23%, *p* = 0.009). The FNCLCC/FFCD trial had a similar 5-year OS (38% versus 24%, *p* = 0.02) without utilizing anthracyclines (epirubicin). Building on this successful approach, the German AIO-FLOT4 trial compared 5-FU plus leucovorin, oxaliplatin, and docetaxel (FLOT) to ECF/X. Patients in the intervention arm were planned to receive four cycles preoperatively and four cycles post-operatively. Median overall survival in the FLOT arm was 50 months versus 35 months, with an estimated 5-year OS rate of 45% versus 36% [27]. It was also associated with a higher pCR rate (16% versus 6%, *p* = 0.02) and an R0 resection rate (84% versus 77%, *p* = 0.01) [9]. Additionally, there was no significant difference in perioperative mortality or complications. Given the superior efficacy outcomes and similar safety profile FLOT triplet chemotherapy was established as a new standard of care for perioperative therapy.

The phase III TOPGEAR study examining the addition of chemoradiation to standard perioperative chemotherapy, initially ECF and since 2017 FLOT as its perioperative regimen, completed accrual in 2021 and enrolled 574 patients for patients with resectable gastric cancer [28]. Interim results demonstrated that preoperative chemoradiation added to perioperative chemotherapy was safe and feasible; however, we await complete efficacy and safety analysis. While it is widely acknowledged that this perioperative approach offers a distinct survival advantage over surgery alone, there are significant concerns regarding the attrition and the low completion rates of adjuvant therapy. Additionally, the above landmark studies were designed to continue the same treatment in the adjuvant setting regardless of whether there were favorable or unfavorable pathologic responses at the time of surgery.

As for distal esophageal or GEJ adenocarcinoma, the currently available clinical data support equipoise between trimodality and perioperative approaches. The Neo-AEGIS and ESOPEC trials were designed to address this question and directly compare the two distinct approaches. Neo-AEGIS randomized 377 patients to either CROSS or perioperative chemotherapy (ECF/FLOT). Despite increased local control outcomes with CROSS (improved R0 resection (96% versus 82%, *p* = 0.0003), pCR (12% versus 4, *p* = 0.012) and MPR (39% versus 12%, *p* ≤ 0.0001)), DFS however favored perioperative chemotherapy with a median DFS of 32.4 months versus 24 months (*p* = 0.41) and 3-year OS estimates were equivocal between the arms (CROSS 57% versus ECF/FLOT 55%, HR 1.03). Due to slow accrual and changes in established standards of care over the enrollment period, only 15 percent of patients received FLOT perioperatively [29,30]. The ESOPEC study was initiated in the FLOT era and is expected to report initial results in the near future.

## 3. Rationale for Using Immunotherapy in Resectable EGC

The addition of immune checkpoint inhibition (ICI) to couplet chemotherapy (FOLFOX/CAPOX) was incorporated into the NCCN guidelines following the seminal trials in the metastatic EGC setting, KEYNOTE 590, CheckMate 648, and CheckMate 649. However, post-hoc analyses have found the magnitude of benefit and efficacy is limited to those harboring elevated PD-L1 expression, particularly combined positive score (CPS) of greater than or equal to 5 and 10, suggesting that patients should be stratified prior to initiation to determine the anticipated degree of response [31,32,33].

CheckMate 577, using adjuvant immunotherapy, namely nivolumab post-neoadjuvant CRT, included both adenocarcinoma and squamous cell carcinoma cohorts and demonstrated significant improvements in its primary endpoint. The median DFS was 22.4 months in the nivolumab arm compared to 11.0 months in the placebo arm (HR 0.69; *p* ≤ 0.001) without significant toxicity or deterioration in quality-of-life scores. As expected, the magnitude of benefit was more pronounced amongst the squamous cell carcinoma population than for the adenocarcinoma population (DFS 29.7 months (CI: 14.4—not reached) versus 19.4 months (CI: 15.9–29.4)). An exploratory post-hoc analysis again demonstrated similar results to those of the metastatic cohorts as patients with elevated baseline CPS scores (≥5) derived the greatest benefit from adjuvant immunotherapy (DFS 29.4 months versus 16.3 months) [25]. As the study enrolled patients irrespective of PD-L1 expression levels, the FDA and EMA approved adjuvant nivolumab biomarker independent. Active surveillance is recommended for those who achieve a pCR. In contrast to esophageal and GEJ cancer, the utility of adjuvant immunotherapy in resected gastric cancer has not been established.

The ATTRACTION-5 phase III randomized, placebo-controlled trial evaluated the addition of nivolumab to adjuvant chemotherapy after upfront gastrectomy for gastric cancer. There was no difference in the 3-year relapse-free survival with nivolumab-chemotherapy compared with placebo-chemotherapy (65.3%; HR, 0.90; *p* = 0.4363) in the intent-to-treat analysis, though post-hoc subgroup analysis suggested that tumors with TPS ≥ 1 may derive particular benefit from the addition of nivolumab (HR 0.33, CI 0.14–0.75) [34].

Administration of immunotherapy perioperatively is hypothesized to generate an optimal immune response, leading to improved pathological responses and survival outcomes. The scientific rationale behind this combinatorial effect is that RT or immune-complementary chemotherapy acts synergistically with immunotherapy as they stimulate endogenous immunogenic cell death with subsequent release of tumor-associated neoantigens promoting maturation of antigen-presenting cells (APC) in the tumor microenvironment (TME) producing more polyclonal T-cells and priming responsiveness to immunotherapy to optimize therapeutic response [35,36]. A substantial body of evidence already available in non-EGC cancers from phase III trials strongly indicates that neoadjuvant immunotherapy stands as a viable and rational treatment strategy. In lung cancer, the CheckMate 816 trial, with the addition of nivolumab to chemotherapy, led to a notable improvement in pCR rates when compared to chemotherapy alone (24% vs. 2%). Importantly, this improvement was achieved without any significant rise in overall toxicity or disruptions to the surgical schedule [12]. The recent phase II study in melanoma comparing neoadjuvant (3 doses) and adjuvant pembrolizumab to adjuvant alone demonstrated an improvement in event-free survival (EFS) at 2 years (72% (CI: 64–80) versus 49% (CI: 41–59)). It’s worthwhile noting that 8% (12/154) had progression prior to resection [13].

## 4. Neoadjuvant and Perioperative Immunotherapy Clinical Trials in EGC

The majority of the reported results exploring the efficacy and safety of immunotherapy are from early phase trials primarily from Asia; however, larger phase III trials are ongoing in European or North American populations. Table 1 and Table 2 represent the majority of clinical trials reported in the neoadjuvant setting for the treatment of esophageal squamous or adenocarcinoma. Many of these studies were single-arm with small sample sizes and included only SCC cohorts. Neoadjuvant chemotherapy without radiation would represent a typical treatment paradigm utilized in Asian populations prior to surgical resection for esophageal cancer. This differs significantly from European or North American norms and should, therefore, not be extrapolated and inferred to Western populations. 

Table 1 and Table 3 summarize the particular characteristics of the selected literature from the published phase I and II studies in esophageal cancer and GEJ/gastric cancer, respectively. Table 2 and Table 4 review the pertinent efficacy and safety endpoints. The tables are divided into those who received immunotherapy with chemoradiotherapy and immunotherapy with chemotherapy with/without VEGF inhibition or immunotherapy alone. We excluded conference abstracts or retrospective cohort studies primarily due to a lack of clarity regarding efficacy or safety data and potential overlap with clinical trial cohorts. The neoadjuvant chemotherapy couplet for esophageal cancer typically included a taxane; docetaxel, paclitaxel or nab-paclitaxel plus platinum; cisplatin, carboplatin or nedaplatin and all patients underwent a minimum of 2 cycles and up to a maximum of 5. Surgical resection typically took place 4–6 weeks post cessation of treatment. The immunotherapy given was primarily a PD-(L)1 inhibitor. To report the outcomes uniformly, we used an intention-to-treat (ITT) model of the full analysis set with the pCR rate and MPR rate defined as the number of patients with pCR and MPR divided by total evaluable patients, those that were initially enrolled. All patients who signed consent and began protocol treatment were considered evaluable. Reporting in a uniform and reproducible manner allows for valid cross-study comparisons. Many of the reported pCR and MPR rates use modified ITT reporting outcomes from patients who were resected, which can lead to selection bias and can potentially over-inflate the efficacy of the data, particularly if resection rates were lower than anticipated.

## 5. Efficacy and Safety of Neoadjuvant Immunotherapy in Esophageal Cancer

Neoadjuvant immunotherapy has shown encouraging efficacy outcomes in individuals diagnosed with esophageal cancer. All the selected studies reported pCR, and 15 of the 16 trials reported the MPR rate. The pooled pCR rate was 29.5% (95% CI, 24.4–34.6%), ranging from 10% to 50% for the 482 patients included in the aforementioned 16 selected trials. The pooled MPR rate from 15 trials was 50% (95% CI, 41.1–58.9%), ranging from 25% to 80%. Safety outcomes such as serious treatment-related adverse events (TRAE grade 3–5) were reported in 15 trials with a pooled rate of 30% (95% CI, 21–39%), although in the two studies that added immunotherapy to CRT, the incidence of serious AEs was 65.0% and 43%, respectively which aligns with historical data. Serious immune-related adverse events (irAE Grade 3–5) were consistently extremely low in the reporting 13 trials. The pooled resection rate was 84% (95% CI, 78.9–89.1%), ranging from 60% to 100%, and the R0 resection rate was similar at 83% (95% CI, 77.2–88.8%) with a range from 60% to 93% indicating that if a patient was successful in reaching surgery, they were likely to have a R0 resection. Several reasons were cited for the decision not to proceed with resection: disease progression, patient refusal, mortality, TRAE, compromised overall health, and dropouts. There were very few treatment-related surgical delays, with only 2 studies reporting delays at 16% and 17%, respectively. Of note, Wang et al. included VEGF inhibitors which could have contributed to the increased toxicity and, therefore, delay [52]. The PERFECT trial had slightly higher rates of anastomotic fistula and chylothorax at 21.2% and 15.2%, respectively, and the median pooled rate of anastomotic fistula and chylothorax in the chemotherapy/immunotherapy trials was 10% (95% CI, 6.4–13.6%) and 5.85% (95% CI, 3.52–8.18%), respectively [38].

## 6. Safety and Efficacy of Neoadjuvant and Perioperative Immunotherapy in GEJ/Gastric Cancer

All seven selected studies reported pCR, R0 resection rate, toxicity and six of the seven trials reported the MPR rate. Excluding the MSI-H study by Andre et al., the pooled rates of pCR and MPR using neoadjuvant immunotherapy were 21% (95% CI, 12–30%) and 47% (95% CI, 28.7–65.3%), respectively. The pooled resection rate was 92% (95% CI, 86.3–97.7%) and the R0 resection rate was 90% (95% CI, 84.7–95%). There were very few treatment-related surgical delays, with only two studies reporting delays at 11% and 17% respectively. TRAE grade 3–5 and serious immune-related events were reported in the 7 trials with a pooled rate of 29% (95% CI, 11–47%) and 13% (95% CI, 7–19%), respectively, although in the two studies that added immunotherapy to CRT, the incidence of serious AEs was 55% and 78%, respectively, which aligns with historical data. Immunotherapy alone demonstrated limited efficacy with a pCR rate of 3% and MPR of 16%, which was primarily derived from MSI-H patients (N = 4).

## 7. Upcoming Directions in Neoadjuvant and Perioperative Immunotherapy for EGC 

Currently enrolling and upcoming trials in neoadjuvant and perioperative mmunotherapy for EGC are noted in Table 5. As many EGC patients are unresectable at diagnosis, KEYNOTE-975 is currently assessing the potential of pembrolizumab in conjunction with definitive chemoradiation for individuals with locally advanced esophageal and GEJ cancer [60]. KUNLUN is a similar phase III trial that aims to evaluate the effectiveness of durvalumab when administered concurrently with and subsequent to definitive chemoradiation; however, it is solely examining esophageal SCC [61]. SKYSCRAPER-07 is a randomized controlled phase III trial that will investigate consolidation atezolizumab, as well as dual inhibition with atezolizumab and tiragolumab, a TIGIT antibody, in unresectable esophageal SCC patients, whose cancers have not progressed following definitive chemoradiotherapy.

The EA2174 (ECOG-ACRIN Cancer Research Group) US study is a randomized phase II/III for patients with locoregional esophageal or GEJ cancer that is deemed resectable evaluating immunotherapy in both the neoadjuvant and adjuvant setting in combination with nCRT and delivering adjuvant immunotherapy with either nivolumab or ipilimumab/nivolumab regardless of pCR status [63]. NCT05357846 and NCT05244798 are similar phase III trials using sintilimab; however, NCT05244798 has a 3rd arm that does not include radiotherapy and will give greater insight into the utility of RT in this Asian population. NICE-2 has a similar three-arm design with similar cohorts [64]. Keystone-002 examines pembrolizumab in combination with nCT and compares it to standard nCRT. KEYNOTE-585 and MATTERHORN are similar phase III studies of perioperative 5FU cisplatin or FLOT with or without pembrolizumab or durvalumab, respectively, for gastric cancer. They have similar expected sample sizes and also similar endpoints [66,67]. Early indications from interim analysis of KEYNOTE-585 indicate that immunotherapy improves pathological regression and demonstrated a statistically significant improvement in pCR rates, 12.9% with pembrolizumab plus chemotherapy versus 2.0% with placebo plus chemotherapy (treatment difference 10.9%; 95% CI 7.5–14.8; *p* < 0.00001); however, there was no statistically significant improvement in EFS (44.4 months versus 25.3 months; HR 0.81; 95% CI 0.67–0.99; *p* = 0.0198) per the prespecified statistical analysis plan [71]. Since the majority of patients in KEYNOTE-585 received fluorouracil plus cisplatin rather than FLOT, survival outcomes from more contemporary studies such as MATTERHORN are eagerly awaited. Interim results from the MATTERHORN study have demonstrated an improvement in pCR (19% versus 7%, respectively; *p* < 0.00001); however, survival outcomes are pending [72]. NCT03044613 was the only study that used a LAG-3 inhibitor in combination with PD-1 inhibition; however, there was significant grade 3 irAE (6/9), two of which were pericarditis, when combined with CRT and led to a protocol amendment and change in treatment sequence with combination moving to induction prior to CRT [62]. DANTE, a randomized, bi-national, phase II study evaluating FLOT with atezolizumab released interim data at ASCO 2022 demonstrating an improvement in pCR rate (24% vs. 15%) and MPR rate (48% vs. 39%) compared to chemotherapy alone. Additionally, the magnitude of response was correlated with increasing levels of PD-L1 CPS score. Four deaths were reported among patients in the experimental arm (3%) [65]. Survival data has yet to be published, but given the early findings from KEYNOTE-585, improvements in pCR may not translate to improved outcomes, and more research is needed to understand this phenomenon. Although the phase II ICONIC study, evaluating perioperative FLOT with avelumab, demonstrated a reasonable safety profile, it was closed early due to futility as it failed to demonstrate an improvement in pCR as the 34 patients only had a pCR of 15 percent [68].

## 8. Microsatellite Instability Defines a Unique Immunotherapy Subpopulation

Microsatellite instability-high (MSI-H) cancers represent a distinct subset of tumors characterized by genetic alterations that affect the stability of repetitive DNA sequences known as microsatellites. This instability arises due to impaired DNA mismatch repair (dMMR) mechanisms, which normally correct errors during DNA replication. As a result, MSI-H cancers accumulate mutations throughout the genome and exhibit unique biological and clinical features such as an inflamed TME.

MSI-H status has gained prominence as both a predictive and prognostic biomarker in various cancer types, influencing treatment strategies and patient outcomes. These tumors tend to have distinct histopathological characteristics and can display increased immunogenicity, leading to enhanced immune responses. They represent approximately 8–10 percent of EGC patients, and several studies have shown that they respond exceptionally well to immunotherapy both in the metastatic and neoadjuvant setting. The MSI-H patients in the DANTE phase II trial experimental arm demonstrated improved pCR (50% versus 27%) and MPR (70% versus 27%) rates compared to standard FLOT chemotherapy [65]. Findings from the French phase II NEONIPIGA study, which investigated neoadjuvant nivolumab and ipilimumab followed by adjuvant nivolumab with resectable MSI-H/dMMR GEJ and gastric cancer, revealed a remarkable pCR rate of 53 percent and MPR rate of 66 percent. Additionally, the patients (two declined surgery, and one had metastasis at inclusion) where resection was deferred had complete endoscopic responses with tumor-free biopsies and normal imaging. The toxicity profile was higher due to the addition of CTLA-4 inhibition but not excessive at only 19 percent due to only 2 doses being administered [55]. The tumor-agnostic French IMHOTEP phase II study enrolled localized resectable dMMR/MSI patients and delivered 1–2 doses of neoadjuvant pembrolizumab prior to resection. It enrolled 21 GEJ patients and reported a pCR of 25 percent amongst the GEJ cohort, although it must be noted that this is in a mITT population, as 23 percent did not proceed to surgery, primarily due to patient preference due to complete clinical response and improvement in swallow function [70]. INFINITY, an Italian phase II, exploratory, proof-of-concept, multi-cohort, single-arm, de-escalation study is investigating neoadjuvant tremelimumab and durvalumab, with cohort 1 undergoing standard surgical resection and if all endpoints are met cohort 2 will undergo the same neoadjuvant therapy however those patients demonstrating a complete clinical response will undergo a non-operative approach and active surveillance [69]. Initial reported data indicate a 60 percent pCR rate and 80 percent MPR rate, and it will likely continue as planned to cohort 2. This study has the potential to deliver practice-changing results. It follows a clear scientific rationale and may avoid considerable perioperative morbidity and mortality, improving long-term quality of life outcomes in this prespecified cohort.

## 9. Conclusions

Neoadjuvant therapy presents an opportune timeframe for translational and clinical evaluation of tumor biology. Unlike adjuvant therapy, the utilization of immunotherapy in the neoadjuvant setting enables in vivo/ex vivo observations of the tumor microenvironment and related immune editing across various immune compartments and at different time points. This can be accomplished by longitudinal translational analysis and examination of sequential liquid and tissue biopsies and, ultimately, the surgical specimen. Such an approach allows for a comprehensive assessment of the treatment’s impact on the tumor’s immune response, providing valuable insights for further research into predictive biomarkers and potential therapeutic advancements. Clinically, tumor response or lack of response to neoadjuvant therapy (as assessed by size and/or metabolic radiographic criteria) can help inform suitability for surgical resection as well as potential additional systemic therapy in the post-operative or recurrent metastatic setting.

Early preliminary positive results from single-arm phase I and II trials indicate that neoadjuvant immunotherapy, either in combination with chemotherapy or chemoradiotherapy, is safe and feasible in EGC with similar resection rates and R0 rates to historical norms with a tolerable and manageable toxicity profile. Ongoing global confirmatory phase II and III trials will further validate the feasibility and safety of this approach and provide critical data about the potential survival benefit, as well as important insights about differences between neoadjuvant and perioperative immunotherapy strategies. Correlative analyses from these large clinical studies will herald the development of biomarkers that can identify patients most likely to benefit from immunotherapy. These predictive biomarkers will be key to selecting appropriate populations for future study and could further improve the personalization of treatment. One could foresee future trials being designed based on a refined risk stratification, with escalation or de-escalation of treatment, in terms of both duration and drug selection, based on pathological response, minimal residual disease status, and genomic and transcriptomic expression changes, amongst others.

Although pathological regression, including pCR, has been validated and become a standard surrogate biomarker endpoint for survival outcomes in other solid tumors, its clinical utility in EGC remains unclear. As shown by CALGB 8083, Neo-AEGIS, and KEYNOTE-585, improvement in pCR may not translate to improvement in event-free survival [29,71,73]. Notably, in KEYNOTE-585, the magnitude of survival benefit was correlated with PD-L1 positivity (HR 0.70 for CPS ≥ 10), suggesting that biomarker-directed patient selection will likely be important for demonstrating meaningful clinical benefit from neoadjuvant immunotherapy in EGC.

Another important question relates to whether prolonged immunotherapy as deployed in the perioperative approach is necessary to improve survival, given potential toxicity and cost-effectiveness concerns. Overtreatment, exposing patients to toxic and potentially futile treatment, remains a significant concern for clinicians, particularly for patients who have demonstrated significant pathological regression. Additionally, one could argue that patients who have had limited pathological responses are being under-treated by pursuing the same treatment post-operatively as it may be similarly futile. Unfortunately, current studies are not designed to answer these questions, highlighting the importance of investigator-initiated studies in answering these questions in the future. The only available evidence for utilization of immunotherapy comparing the various approaches in the perioperative setting is in NSCLC. Multiple analyses have found that a neoadjuvant approach is more cost-effective than an adjuvant approach [74,75]. In contrast, a perioperative approach, including four neoadjuvant and 13 adjuvant doses of immunotherapy, is the least cost-effective at an incremental cost–effectiveness ratio (ICER) of $94,222.29/QALY, nearly three times that of neoadjuvant [76]. To give context, in the EGC setting, the ICER for adjuvant nivolumab based on CheckMate 577 is $42,733/QALY ($71,474 for 1.67 QALY gained) [77].

As our knowledge of the heterogeneity of EGC deepens and our therapies evolve, we must refine and tailor previous historical treatment standards to ensure optimal, efficient utilization. This is exemplified by total neoadjuvant therapy (TNT), as it has transformed the treatment landscape of lower GI cancers based on the phase III RAPIDO and UNICANCER-PRODIGE 23 studies. One could argue for the inclusion of short-course radiotherapy in future trial designs for EGC, given the potential synergy between radiation, chemotherapy, and immunotherapy. This would attempt to bridge the divide between improved local control and long-term distant control that may be missing with low-dose chemo-sensitizing agents [78,79].

While open questions remain about optimizing immunotherapy in resectable EGC, including timing and dosing in the perioperative setting and the potential synergistic impact of radiotherapy, we are at the dawning of a new era in EGC management that will accelerate our understanding of immune interactions in EGC and ultimately improve patient outcomes globally.

## Figures and Tables

**Table 1 cancers-16-00286-t001:** Selected neoadjuvant trials administering immunotherapy or chemo-immunotherapy for patients with resectable EC.

Study	Year	Name/NCT Number	Study Type	Country	Sample Size (N)	Stage/Path	Treatment	Primary Endpoint
**Neoadjuvant Immunotherapy with Chemoradiotherapy**								
Li, C.Q. et al. [37]	2021	PALACE-1	Phase 1b	China	20	Stage II–IVA ESCC	Pembrolizumab Carboplatin/paclitaxel	Safety
van den Ende T [38]	2021	PERFECT	Phase 2	Netherlands	40	Stage I–IVA EAC	Atezolizumab Carboplatin/paclitaxel	feasibility
**Neoadjuvant Immunotherapy with Chemotherapy**								
Shen et al. [39]	2021	NR	Phase 2	China	28	Stage II–IVA ESCC	Nivolumab or pembrolizumab or camrelizumab Carboplatin/Nab-paclitaxel	Safety, feasibility
Zhang, Z.Y et al. [40]	2021	ESONICT-1	Phase 2	China	30	Stage II–IV ESCC	Sintilimab Cisplatin/Nab-paclitaxel	pCR, Safety
Duan et al. [41]	2021	SIN-ICE	Pilot	China	23	Stage II–IVA ESCC	Sintilimab Nedaplatin/Nab-paclitaxel or docetaxel	pCR, safety
Peng Yang et al. [42]	2021	ChiCTR2100051903	Pilot	China	16	Stage II–IVA ESCC	Camrelizumab Carboplatin/Nab-paclitaxel	pCR
Xing et al. [43]	2021	NCT 03985670	Phase 2	China	30	Stage II–IVA ESCC	Toripalimab Cisplatin/paclitaxel	pCR
Yang, W.X. et al. [44]	2022	ChiCTR2000028900	Pilot	China	23	Stage II–III ESCC	Camrelizumab Carboplatin/Nab-paclitaxel	Safety, feasibility
He et al. [45]	2022	NCT 04177797	Phase 2	China	20	Stage III–IVA ESCC	Toripalimab Carboplatin/paclitaxel	Safety, feasibility, MPR
Liu et al. [46]	2022	NICE	Phase 2	China	60	Stage III–IVA ESCC	Camrelizumab Carboplatin/Nab-paclitaxel	pCR
Gao et al. [47]	2022	ESONICT-2	Phase 2	China	20	Stage III–IVA ESCC	Toripalimab Cisplatin/docetaxel	pCR, AEs
Jun Liu et al. [48]	2022	NIC-ESCC2019	Phase 2	China	56	Stage II–IVA ESCC	Camrelizumab Cisplatin/Nab-paclitaxel	pCR
Duan et al. [49]	2022	PEN-ICE	Phase 2	China	18	Stage II–IVA ESCC	Pembrolizumab Nedaplatin/Nab-paclitaxel or docetaxel	Safety, efficacy
Yan et al. [50]	2022	TD-NICE	Phase 2	China	45	Stage II–IVA ESCC	Tislelizumab Carboplatin/Nab-paclitaxel	MPR
Huang et al. [51]	2021	ChiCTR2000035079	Phase 2	China	23	Stage II–IVA ESCC	Pembrolizumab Nedaplatin/docetaxel	pCR
**Neoadjuvant Immunotherapy with Chemotherapy and VEGF Inhibitor**								
Wang et al. [52]	2022	ChiCTR1900023880	Phase 1b	China	30	Stage II–III ESCC	Camrelizumab Apatinib Nedaplatin/Nab-paclitaxel	Safety

Abbreviations: N—number of patients; pCR—pathological complete remission; MPR—major pathological response; AE—Adverse events; NR—not recorded; NCT—National Clinical Trial; ESCC—esophageal squamous cell carcinoma; EAC—esophageal adenocarcinoma.

**Table 2 cancers-16-00286-t002:** Surgical feasibility, safety and efficacy outcomes for selected trials in EC.

Study	Sample Size	Pathology	Surgical Resection Rate N (%)	Surgical Delay Rate N (%)	RO Resection Rate N (%)	Incidence of 3–5 TRAE-%	Grade ≥ 3 ir-AEs-%	MPR/TRG1–2 Rates-%	pCR Rates-%
**Neoadjuvant Immunotherapy with Chemoradiotherapy**									
Li, C.Q. et al. [37]	20	ESCC	18 (90)	1 (6)	17 (85)	65%	NR	80 (16/20)	50 (10/20)
van den Ende T [38]	40	EAC	33 (83)	0 (0)	33 (83)	43%	5%	33 (13/40)	25 (10/40)
**Neoadjuvant Immunotherapy with Chemotherapy**									
Shen et al. [39]	28	ESCC	27 (96)	0 (0)	26 (93)	7.1%	3.5%	79 (22/28)	32 (9/28)
Zhang, Z.Y et al. [40]	30	ESCC	23 (77)	0 (0)	23 (77)	3%	3%	40 (12/30)	13 (4/30)
Duan et al. [41]	23	ESCC	17 (74)	0 (0)	16 (70)	30.4%	0%	39 (9/23)	26 (6/23)
Peng Yang et al. [42]	16	ESCC	16 (100)	NR	15 (94)	NR	NR	81 (13/16)	31 (5/16)
Xing et al. [43]	30	ESCC	24 (80)	NR	24 (80)	30%	3.3%	NR	17 (5/30)
Yang, W.X. et al. [44]	23	ESCC	20 (87)	0 (0)	20 (87)	39%	0%	43 (10/23)	22 (5/23)
He et al. [45]	20	ESCC	16 (80)	0 (0)	14 (70)	22%	NR	35 (7/20)	15 (3/20)
Liu et al. [46]	60	ESCC	51 (85)	8 (16)	50 (83)	56.7%	5%	59 (35/60)	33 (20/60)
Gao et al. [47]	20	ESCC	12 (60)	0 (0)	12 (60)	20%	5%	25 (5/20)	10 (2/20)
Jun Liu et al. [48]	56	ESCC	51 (91)	0 (0)	51 (91)	11%	3.6%	54 (30/56)	29 (16/56)
Duan et al. [49]	18	ESCC	13 (72)	0 (0)	11 (61)	28%	0%	50 (9/18)	33 (6/18)
Yan et al. [50]	45	ESCC	36 (80)	0 (0)	29 (64)	42%	0%	58 (26/45)	40 (18/45)
Huang et al. [51]	23	ESCC	21 (91)	0 (0)	21 (91)	13%	0%	43 (10/23)	30 (7/23)
**Neoadjuvant Immunotherapy with Chemotherapy and VEGF inhibitor**									
Wang et al. [52]	30	ESCC	29 (97)	5 (17)	28 (93)	37%	13%	50 (15/30)	35 (7/20)

Abbreviations: N—number of patients; %—percentage; R0—complete resection with clear margins; TRAE—treatment-related adverse events; ir-AE—immune-related adverse events; pCR—pathological complete remission; MPR—major pathological response; TRG—Tumor regression grade; NR—not recorded; ESCC—esophageal squamous cell carcinoma; EAC—esophageal adenocarcinoma.

**Table 3 cancers-16-00286-t003:** Selected neoadjuvant and perioperative trials administering immunotherapy or chemo-immunotherapy for patients with resectable GEJ and gastric cancer.

Study	Year	Name/NCT Number	Study Type	Country	Sample Size (N)	Stage/Path	Treatment	Primary Endpoint
**Neoadjuvant Immunotherapy with Chemoradiotherapy**								
Zhu et al. [53]	2022	MC1541	Phase Ib/2 trial	USA	31	Stage I–IIIB AC GEJ	Pembrolizumab + Carboplatin/paclitaxel	Safety, feasibility, pCR
Z. Tang et al. [54]	2022	Neo-PLANET	Phase 2	China	36	Stage I–IIIC AC GEJ/GC	Camrelizumab + CAPOX +capecitabine/RT	pCR
**Neoadjuvant Immunotherapy with Chemotherapy**								
Andre et al. [55]	2023	GERCOR NEONIPIGA	phase II	France	32	Stage I–IIIB AC GEJ/GC-dMMR/MSI-H	Nivolumab, Ipilimumab	pCR, AEs
Sun et al. [56]	2023	NCT03488667	Phase 2	USA	37	Stage I–IVA AC EC/GEJ/GC	Pembrolizumab + FOLFOX	pCR, Safety, feasibility
Jiang H et al. [57]	2022	NCT04065282	Phase 2	China	36	Stage I–IIIC AC GEJ/GC	Sintilimab + CAPOX	pCR
Guo H. et al. [58]	2022	ChiCTR2000030414	Phase 2	China	30	Stage I–IIIC AC GC	Sintilimab + CAPOX	pCR
**Neoadjuvant Immunotherapy Alone**								
Hasegawa et al. [59]	2022	ONO-4538-67	Phase 1	Japan	31	Stage I–IIIC AC GC	nivolumab	Safety

Abbreviations: N—number of patients; AE—adverse events; NCT—National Clinical Trial; pCR—pathological complete remission;; AC GEJ/GC—adenocarcinoma of the gastroesophageal junction or gastric adenocarcinoma, EC—esophageal adenocarcinoma; CAPOX—capecitabine/oxaliplatin; FOLFOX—5FU/oxaliplatin; dMMR/MSI-H—mismatch repair deficiency/microsatellite instability-high; RT—radiotherapy.

**Table 4 cancers-16-00286-t004:** Surgical feasibility, safety and efficacy outcomes for selected trials in GEJ and gastric cancer.

Study	Sample Size	Pathology	Surgical Resection Rate N (%)	Surgical Delay Rate N (%)	RO Resection Rate N (%)	Incidence of 3–5 TRAE-%	Grade ≥ 3 ir-AEs-%	MPR/TRG1–2 Rates-%	pCR Rates-%
**Neoadjuvant Immunotherapy with Chemoradiotherapy**									
Zhu et al. [53]	31	AC	28 (90)	0 (0)	28 (90)	55%	13%	NR	23 (7/31)
Z. Tang et al. [54]	36	AC	33 (92)	6 (17)	33 (92)	78%	14%	44 (16/36)	33 (12/36)
**Neoadjuvant Immunotherapy with chemotherapy**									
Andre et al. [55]	32	AC dMMR/MSI-H	29 (90)	0 (0)	29 (90)	19%	19%	66 (21/31)	53 (17/31)
Sun et al. [56]	37	AC EC/GEJ/GC	29 (78)	0 (0)	29 (78)	61%	10%	70 (26/37)	16 (6/37)
Jiang H et al. [57]	36	AC GEJ/GC	36 (100)	4(11)	35 (97)	28%	0%	47 (17/36)	19 (7/36)
Guo H. et al. [58]	30	AC GC	30 (100)	0 (0)	30 (100)	10%	10%	63 (19/30)	33 (10/30
**Neoadjuvant immunotherapy alone**									
Hasegawa et al. [59]	31	AC GC	30 (97)	0 (0)	27 (87)	29%	29%	16 (5/31)	3 (1/31)

Abbreviations: N—number of patients; %—percentage; R0—complete resection with clear margins; TRAE—treatment-related adverse events; ir-AE—immune-related adverse events; pCR—pathological complete remission; MPR—major pathological response; TRG—Tumor regression grade; NR—not recorded; AC—adenocarcinoma; dMMR/MSI-H—mismatch repair deficiency/microsatellite instability-high; EC—esophageal adenocarcinoma; GEJ—gastroesophageal junction adenocarcinoma; GC—gastric adenocarcinoma.

**Table 5 cancers-16-00286-t005:** Ongoing neoadjuvant and perioperative immunotherapy clinical trials in EGC.

Trial Name/ NCT	Study Type	Treatment	Country	Sample Size (N)	Stage/Path	No. of Cycles Immunotherapy	Primary Endpoint
**Perioperative Immunotherapy with definitive chemoradiotherapy**							
KEYNOTE-975 [60]	Phase 3 RCT	Pembrolizumab FOLFOX or Cisplatin/5FU	Global	600	stage I–IVA EC/GEJ	12 months	EFS, OS
KUNLUN [61]	Phase 3 RCT	Durvalumab Cisplatin/5FU or Cisplatin/capecitabine	Global	600	stage II–IVA ESCC	24 months	EFS, OS
SKYSCRAPER-07	Phase 3 RCT	Consolidation Atezolizumab ± Tiragolumab	Global	750	stage II–IVA ESCC	12 months	PFS, OS
**Perioperative Immunotherapy with chemoradiotherapy**							
NCT03044613 [62]	Phase 1b/2	Nivolumab + CP Nivolumab/relatlimab + CP	USA	32	Stage II–IVA ESCC, EC/GEJ	2 cycles induction IO followed by 3 cycles concurrent with chemoRT	Safety
EA2174 [63]	phase 2/3	Neoadjuvant: CP standard Nivolumab + CP Adjuvant: Nivolumab Nivolumab/ipilimumab	USA	278	stage II–III EC/GEJ	2 cycles concurrent Neoadjuvant 13 months adjuvant	pCR
NICE-2 [64]	Phase 2 RCT-3 arm	Camrelizumab + carboplatin/nab-paclitaxel Camrelizumab + CP CP	China	204	stage II–IVA ESCC	2 12 months adjuvant	pCR
NCT05357846	Phase 3 RCT-2 arm	Sintilimab Cisplatin/paclitaxel	China	422	stage II–IVA ESCC	2	OS
NCT05244798	Phase 3 RCT-3 arm	Sintilimab + carboplatin/nab-paclitaxel Sintilimab + carboplatin/nab-paclitaxel + RT carboplatin/nab-paclitaxel + RT	China	420	stage II–IVA ESCC	2	pCR
Keystone-002	Phase 3 RCT-2 arm	Pembrolizumab + cisplatin/paclitaxel cisplatin/paclitaxel + RT	China	342	stage II–IVA ESCC	3 neoadjuvant 12 months adjuvant	EFS
**Perioperative Immunotherapy with chemotherapy**							
DANTE [65]	Phase 2/3 RCT	Atezolizumab + FLOT	Germany, Switzerland	295	stage IB–IIIC AC GEJ/GC	4 neoadjuvant 12 months adjuvant	EFS
KEYNOTE-585 [66]	Phase 3 RCT	Pembrolizumab + FLOT or Cisplatin/5FU or Cisplatin/capecitabine	Global	800	stage IB–IIIC AC GEJ/GC	3 neoadjuvant 12 months adjuvant	OS, EFS, pCR
MATTERHORN [67]	Phase 3 RCT	Durvalumab + FLOT	Global	900	stage II–IIIC AC GEJ/GC	2 cycles neoadjuvant 12 months adjuvant	EFS
ICONIC [68]	Phase 2	Avelumab + FLOT	UK	40	stage I–IIIC AC GEJ/GC	4 neoadjuvant 4 adjuvant	pCR
**Neoadjuvant immunotherapy alone**							
INFINITY [69]	Phase 2	Durvalumab/ Tremelimumab	Italy	31	stage I–IIIB AC GEJ/GC-dMMR/MSI-H	3, 1	pCR
IMHOTEP [70]	Phase 2	Pembrolizumab	France	120	Resectable dMMR/MSI-H	1–2	pCR

Abbreviations: RCT—randomized controlled trial; N—number of patients; NCT—National Clinical Trial; pCR—pathological complete remission; MPR—major pathological response; TRG—Tumor regression grade; AC GEJ/GC—adenocarcinoma of the gastroesophageal junction or gastric adenocarcinoma; CP—carboplatin/paclitaxel; EC—esophageal adenocarcinoma; EFS—event-free survival; OS—overall survival; FOLFOX—5FU/oxaliplatin; pCR—pathological complete remission; dMMR/MSI-H—mismatch repair deficiency/microsatellite instability-high; RT—radiotherapy; ESCC—esophageal squamous cell carcinoma; EAC—esophageal adenocarcinoma; IO—immunotherapy; RT—radiotherapy; FLOT—5FU/oxaliplatin/docetaxel.

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
