# Peer review of "The Evolving Landscape of Neoadjuvant Immunotherapy in Gastroesophageal Cancer"

_cancers, 2024, doi:10.3390/cancers16020286_

Round 1

Reviewer 1 Report

Comments and Suggestions for Authors

The authors have undertaken a significant task in compiling and analyzing data and overall, the manuscript provides valuable insights into neoadjuvant immunotherapy for esophagogastric cancer.

Below are my minor suggestions:

1.       Please check the order of citations in the text. For example, in line 42 it should be 6 instead of 64 and overall adjustment is required throughout the manuscript.

Author Response

We apologize for the oversight. All references and order of citations have been re-checked and updated as needed.

Reviewer 2 Report

Comments and Suggestions for Authors

The review offers a comprehensive and detailed exploration of neoadjuvant and perioperative immunotherapy for EGC. I recommend some adjustments for clarity and emphasis on clinical significance; and how the observed improvements in pathological regression might translate into a better patient outcome. 

Author Response

We would like to thank the reviewer for their suggestion. Although pathological regression, including pCR, has been validated and become a standard surrogate biomarker endpoint for survival outcomes in other solid tumors its clinical utility in EGC remains unclear. As shown by CALGB 8083, Neo-AEGIS, and KEYNOTE-585, improvement in pCR may not translate to improvement in event free survival. Notably, in KEYNOTE-585 the magnitude of survival benefit was correlated with PD-L1 positivity (HR 0.70 for CPS ≥10), suggesting that biomarker-directed patient selection will likely be important for demonstrating meaningful clinical benefit from neoadjuvant immunotherapy in EGC.

Early indications from interim analysis of KEYNOTE-585 indicate that immunotherapy improves pathological regression and demonstrated a statistically significant improvement in pCR rates, 12.9% with pembrolizumab plus chemotherapy versus 2.0% with placebo plus chemotherapy (treatment difference 10.9%; 95% CI 7.5–14.8; p<0.00001); however, there was no statistically significant improvement in EFS (44.4 months versus 25.3 months; HR 0.81; 95% CI 0.67–0.99; p=0.0198) per the pre-specified statistical analysis plan. Since the majority of patients in KEYNOTE-585 received fluorouracil plus cisplatin rather than FLOT, survival outcomes from more contemporary studies such as MATTERHORN are eagerly awaited. Interim results from the MATTERHORN study have demonstrated an improvement in pCR (19% versus 7%, respectively; p<0.00001) however, survival outcomes are pending. We have included the above discussion in the conclusion (lines @399-@406), and the Upcoming directions section expanding on the recent interim analysis from both Keynote 585 and Matterhorn (lines @308-@319) of the revised manuscript.

Reviewer 3 Report

Comments and Suggestions for Authors

this is a comprehensive review of chemo-immunotherapy strategies for esophagogastric cancer - tables are clear and comprehensive with completed and accruing  studies, references 71 and 72 are incomplete

One recurring theme in this area is whether immunotherapy is better pre or post op or both - the cost implications of each strategy is significant - while the authors discuss these issues pertaining to melanoma and lung cancer i feel more detailed  reflection of this issue is warranted - in triple negative breast cancer pembrolizumab is given for a year at a cost of 150,000 dollars regardless of response to initial preop therapy - the current trials represent a watershed where the role of each component can be dissected - as in breast cancer when the regimen is approved the company will not fund a study looking at less therapy (Mousaki I et al. JAMA Network Open 2022 e2216058). The available shorter trials in melanoma and lung cancer were all investigator rather than pharma initiated. 

Author Response

We thank the reviewer for these important suggestions. All references and citations have been re-checked and updated as needed. We agree that the question regarding the appropriateness of a perioperative approach is an important one, particularly in current trial design, with the default stance of many studies to include 12 months of adjuvant therapy. We have included the below discussion in the conclusion and we believe it encapsulates the clinical and financial concerns that have been expressed (lines @407-@423).

Another important question relates to whether prolonged immunotherapy as deployed in the perioperative approach is necessary to improve survival given potential toxicity and cost-effectiveness concerns. Overtreatment, exposing patients to toxic and potentially futile treatment, remains a significant concern for clinicians, particularly for patients who have demonstrated significant pathological regression. Additionally, one could argue that patients who have had limited pathological response are being under-treated by pursuing the same treatment post operatively as it may be similarly futile. Unfortunately, current studies are not designed to answer these questions highlighting the importance of investigator-initiated studies in answering these questions in the future. The only available evidence for utilization of immunotherapy comparing the various approaches in the perioperative setting is in NSCLC. Multiple analyses have found that a neoadjuvant approach is more cost-effective than an adjuvant approach. In contrast a perioperative approach, including 4 neoadjuvant and 13 adjuvant doses of immunotherapy, is the least cost-effective at an incremental cost–effectiveness ratio (ICER) of $94,222.29/QALY, nearly three times that of neoadjuvant. To give context, in the EGC setting the ICER for adjuvant nivolumab based on CheckMate 577 is $42,733/QALY ($71,474 for 1.67 QALY gained).

Reviewer 4 Report

Comments and Suggestions for Authors

Well done

Author Response

Thank you for your review.
